# Bias Field Correction in MRI with Hampel Noise Denoising Diffusion Probabilistic Model

**Junhyeok Lee**[1]                                                201802848@hufs.ac.kr
**Junghwa Kang**[*1]                                            kangjung9592@gmail.com
**Yoonho Nam**[1]                                              yoonhonam@hufs.ac.kr
[1] *Department of Biomedical Engineering, Hankuk University of Foreign Studies, Yongin, KOREA, REPUBLIC of Korea*

**TeaYoung Lee**[2]                                                tylee@pusan.ac.kr
[2] *Department of Neuropsychiatry, Pusan National University Yangsan Hospital, Yangsan, Republic of Korea*

**Editors:** Under Review for MIDL 2023

## Abstract

Non-uniform bias field due to external factors hampers quantitative MR image analysis. For reliable quantitative MR image analysis, appropriate correction for the bias field is necessary. In this study, we propose Hampel denoising diffusion model to effectively correct the bias field from MR images. Compared with N4 and Gaussian denoising diffusion models, the proposed model provided higher PSNRs, SSIMs and lower MSEs. Higher efficiency could be achieved compared to N4 when our model takes 9 times faster in inference time.

**Keywords:** Diffusion model, Bias field, intensity inhomogeneity correction, Magnetic resonance imaging

## 1. Introduction

Magnetic Resonance Imaging (MRI) is a widely used medical imaging modality. But bias field obscure subtle details and impede accurate identification (Meyer et al., 1995; Vovk et al., 2007). N4 (Tustison et al., 2010) has been a commonly used method for correcting the inhomogeneities, however, this method has limitations in terms of its accuracy, technical factor, and efficiency. We propose Hampel Denoising Diffusion Model (HDDnet) conceived to model inhomogeneities by Cauchy-Lorentz distribution (Borgia et al., 1996). We modeled the Hampel mixture distribution to represent the image intensity disrupted by the inhomogeneities. To assess the fitness of Hampel function to the image intensity, the mean fitting error between the histogram and the probability function was calcuated, shown in Figure 1. The intensity difference between the input image and t-step image of diffusion process was used in both histogram and the probability function. The mean fitting error is 0.012 less in Hampel function showing that it is a much better fit to MRI with the bias field, Figure 1A-D (Nachmani et al., 2021). Proposed method effectively corrects the bias field and generates reduced inhomogeneities MRI with higher accuracy and faster inference time than N4.

---

* Contributed equally

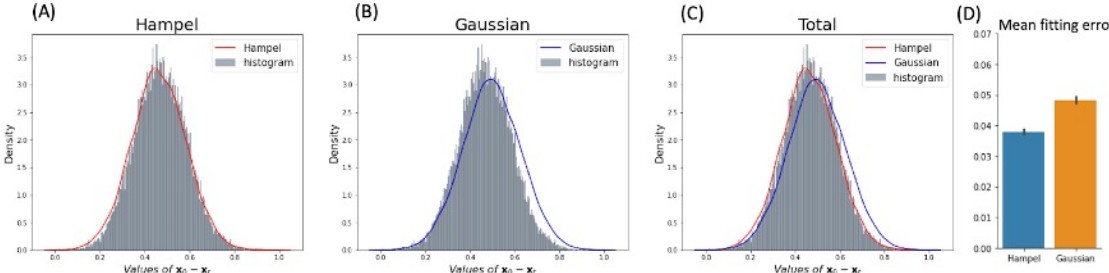

Figure 1: (A) Hampel distribution (B) Gaussian distribution (C) Comparing the fit
(D) The mean fitting MSE between the histogram and the density function

## 2. Method and Experiments

**Method**: We modeled the Hampel Mixture distribution (Hampel and Zurich, 1998) to represent the image intensity disrupted by the inhomogeneities. Denote $\mathbb{H}(\alpha, x_0, \gamma)$ as the Hampel mixture distribution, where $\alpha, x_0, \gamma$ are weight, location, scale parameter, respectively; We use the term $F_h(x, \alpha), F_n(x; 0, 1), F_c(x; x_0, \gamma)$ as probability distribution function of Hampel, Gaussian, and Cauchy-Lorentz, respectively. Hampel function[1] could be written

$$F_h(x, \alpha) = (1 - \alpha)F_n(x; 0, 1) + \alpha F_c(x; x_0, \gamma) \ \ with \ \ 0 \leq \alpha \leq 1$$
$$F_n(x; 0, 1) = \frac{1}{\sqrt{2\pi}}exp(\frac{-x^2}{2}), \ \ F_c(x; x_0, \gamma) = \frac{1}{\pi}\left(\frac{\gamma^2}{(x - x_0)^2 + \gamma^2}\right)$$

Hampel function was optimized with MLE[2] (Haynes, 2013). Through maximizing the Hampel function, we were able to allocate $(\alpha, x_0, \gamma)$ as $(1e - 05, 0.6332, 0.0274)$. Detail explanation and the source code can be found in our Github repository[3].

$$\mathbb{H}(\alpha, x_0, \gamma) = \mathbb{H}(1e - 05, 0.6332, 0.0274)$$

**Dataset**: This study was approved by the Institutional Review Board. We used 202 subjects (126 male, 76 female, age $26.27 \pm 7.84$ years) scanned on a 3T MRI following 3D gradient echo protocol with MT pulse (Nam et al., 2017; Nam et al.). Each of the brain slices is resized to a size of $512 \times 512$ and normalized the values to range between $[0, 1]$. Our dataset is composed of $6,000$ images ($n = 176$) to train and $780$ images ($n = 26$) to test.

**Model and Training**: HDDnet is trained on Nvidia RTX 3090 GPU 24GB with the batch size of 8 for 512 iterations. HDDnet is trained with $L2$ loss, the sigmoid noise schedule for 1,000 steps, a learning rate of $10^{-6}$ for the Adam optimizer, the first layer is chosen as 64.

**Evaluation**: Evaluation was took in both quantitative and qualitative. For the quantitative evaluation, MSE, PSNR, and SSIM[4](Wang et al., 2004) were used. Each was calculated between model output and the N4 label image. Inference time was measured in same

---

1. Hampel mixture probability distribution function

2. Maximum Likelihood Estimation

3. github.com/junhyk-lee/Bias-Field-Correction

4. Mean Squared Error, Peak Signal-to-Noise Ratio, Structural Similarity Index Map, respectively

environment with training setup, with 26 patients. The qualitative assessment was done by comparing N4 and HDDnet prediction of synthetic bias field. The synthetic bias field was generated from train image bias field merged to test set, shown in Figure 2(A).

## 3. Results

As shown in Table 1(A), Hampel random noise outperformed Gaussian random noise in MSE, PSNR, SSIM. Quantitatively, Hampel mixture distribution can provide clear evidence of convergence. Figure 2(B) shows N4 and HDDnet follow similar pattern of the bias field in synthetic image. But as in Table 1(B), our model outper-

|   | Model | MSE | PSNR | SSIM | Time |
|---|-------|-----|------|------|------|
| A | Gaussian | 0.0004 | 32.486 | 0.950 | 4.471 |
|   | Hampel | 0.0003 | 35.945 | 0.983 | 4.473 |
| B | N4 | 0.0003 | 34.766 | 0.979 | 39.601 |
|   | HDD | 0.0001 | 36.865 | 0.978 | 4.478 |

Table 1: Evaluation Metrics

formed on its MSE and PSNR, while SSIM is small in difference. While N4 takes average of 39.6014 secs to correct the bias field of from its corrupted MRI, HDDnet takes about average of 4 secs, which is 9.75 times faster. While maintaining or improving the bias field correction our model shows high efficiency in time.

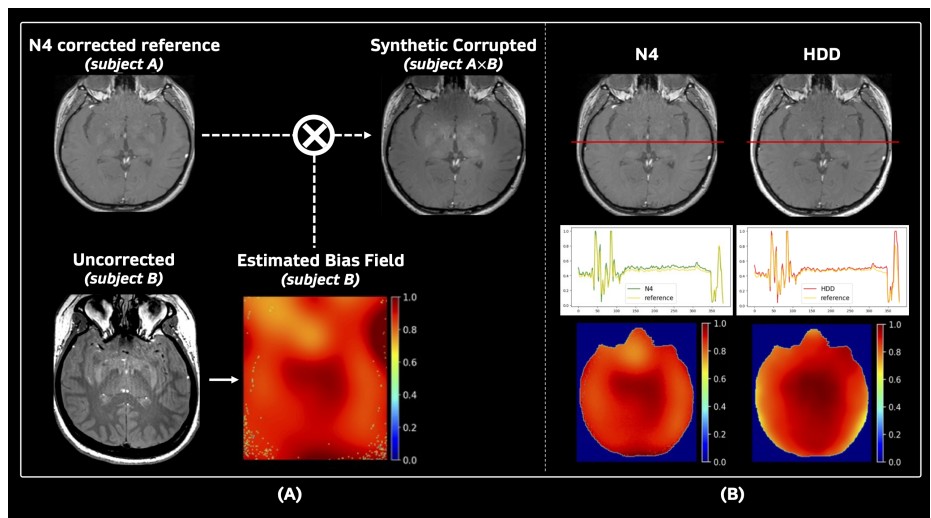

Figure 2: Comparison of bias field estimation results with synthetic bias field

## 4. Conclusion

In this paper, we propose a new bias field correction method by altering the Gaussian noise to Hampel noise, a mixture of Gaussian distribution and Cauchy-Lorentz distribution. Our proposed method is more robust with automatic parameter settings on correcting the bias field than N4. Such automation can give less complexity to the user. We also point out that such deep learning approach is faster in time while still maintaining the accuracy.

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
