# OpenReview forum: "Bias Field Correction in MRI with Hampel Noise Denoising Diffusion Probabilistic Model"
_MIDL.io/2023/Short_Paper_Track — MIDL 2023 Short paper track Poster_

### Official Review · Reviewer_P7Pb · 2023-04-24
**Paper 121 review**

**Rating:** 6
**Confidence:** 2

**Review:**

This paper proposed to use a Hampel denoising diffusion model to correct the bias field in MR images. The method is compared to using a normal distribution; however, there are many methods for correcting MR Bias fields which does not seem to be mentioned in this paper. It is not clear what advantages the proposed method would have over established methods.

---

### Official Review · Reviewer_jmCF · 2023-04-24
**Bias Field Correction in MRI with Hampel Noise Denoising Diffusion Probabilistic Model**

**Rating:** 7
**Confidence:** 3

**Review:**

The article proposes a Hampel denoising diffusion model to correct the bias field in magnetic resonance imaging. The authors compared their model with N4 and Gaussian denoising diffusion models and found that their model provided higher PSNRs, SSIMs, and lower MSEs, with higher efficiency compared to N4. The authors also demonstrated the fitness of the Hampel function to MRI with the bias field and showed that their method effectively corrected the bias field and generated reduced inhomogeneities MRI with higher accuracy and faster inference time than N4.

The article is written in clear and concise language, and the proposed method is presented in a well-structured manner. The authors provide a thorough description of the methodology used in their research, including the dataset, model, and evaluation. The figures and tables included in the article are also informative and easy to read.

The originality and significance of the research lie in proposing a new method for correcting the bias field in MRI, which is an essential step for reliable quantitative MR image analysis. The authors compared their model with two other methods and showed that their model provided better results. The article also highlights the limitations of N4 in terms of accuracy, technical factors, and efficiency, which justifies the need for alternative methods.

While the proposed method appears to show promising results, there are several limitations that should be considered. The study did not evaluate the impact of noise on the proposed method's performance. It is unclear how well the proposed method will perform on noisy images.

Overall, the article provides a valuable contribution to the field of medical imaging by proposing a new method for correcting the bias field in MRI. The article is well-written, and the proposed method shows promise in improving the accuracy and efficiency of quantitative MR image analysis.